# Anti Gram-Positive Bacteria Activity of Synthetic Quaternary Ammonium Lipid and Its Precursor Phosphonium Salt

**DOI:** 10.3390/ijms25052761

**Published:** 2024-02-27

**Authors:** Francesca Bacchetti, Anna Maria Schito, Marco Milanese, Sara Castellaro, Silvana Alfei

**Affiliations:** 1Department of Pharmacy, University of Genoa, 16148 Genoa, Italy; francesca.bacchetti@edu.unige.it (F.B.); marco.milanese@unige.it (M.M.); sara.caste@libero.it (S.C.); 2Department of Surgical Sciences and Integrated Diagnostics (DISC), University of Genoa, 16132 Genoa, Italy; amschito@unige.it; 3IRCCS Ospedale Policlinico San Martino, 16132 Genoa, Italy

**Keywords:** multi-drug-resistant bacteria, quaternary ammonium lipid (**6**), phosphonium salt (**1**), membrane permeabilization, Gram-positive and Gram-negative bacteria, MICs determination, time-killing experiments, cytotoxicity studies

## Abstract

Organic ammonium and phosphonium salts exert excellent antimicrobial effects by interacting lethally with bacterial membranes. Particularly, quaternary ammonium lipids have demonstrated efficiency both as gene vectors and antibacterial agents. Here, aiming at finding new antibacterial devices belonging to both classes, we prepared a water-soluble quaternary ammonium lipid (**6**) and a phosphonium salt (**1**) by designing a synthetic path where **1** would be an intermediate to achieve **6**. All synthesized compounds were characterized by Fourier-transform infrared spectroscopy and Nuclear Magnetic Resonance. Additionally, potentiometric titrations of NH_3_^+^ groups **1** and **6** were performed to further confirm their structure by determining their experimental molecular weight. The antibacterial activities of **1** and **6** were assessed first against a selection of multi-drug-resistant clinical isolates of both Gram-positive and Gram-negative species, observing remarkable antibacterial activity of both compounds against Gram-positive isolates of *Enterococcus* and *Staphylococcus* genus. Further investigations on a wider variety of strains of these species confirmed the remarkable antibacterial effects of **1** and **6** (MICs = 4–16 and 4–64 µg/mL, respectively), while 24 h-time-killing experiments carried out with **1** on different *S. aureus* isolates evidenced a bacteriostatic behavior. Moreover, both compounds **1** and **6**, at the lower MIC concentration, did not show significant cytotoxic effects when exposed to HepG2 human hepatic cell lines, paving the way for their potential clinical application.

## 1. Introduction

Quaternary ammonium salts (QASs), such as the well-known benzalkonium chloride, commercially available as CITROSIL^®^ [1], and several of its derivatives, have attracted attention as antimicrobial agents since the 1930s [2]. QASs demonstrated considerable broad-spectrum potency against Gram-positive and Gram-negative bacteria and some fungi [3]. Conventional QASs contain one or more positively charged nitrogen atoms (N^+^), which constitute the cationic head, and which can be inserted either in the aromatic ring of pyridine, imidazole, quinoline, and isoquinoline or in a straight alkyl chain [4]. The N^+^ covalently binds four carbon atoms, of which at least one belongs to a C8–C18 alkyl chain, resembling the chains of fatty acids [4]. QASs are water-soluble due to their hydrophilic ammonium head, which has a chloride (Cl^−^) or bromide (Br^−^) atom as an anion. As an example, QASs comprehend numerous commercially available lipid-based transfection reagents for gene delivery, such as *N*-[1-(2,3-dioleyloxy)propyl]-*N,N,N*-trimethylammonium chloride (DOTMA), 2,3-dioleyloxy-*N*-[2(carboxamidospermine)ethyl]-*N,N*-dimethyl-1-propanaminotrifluoroacetate (DOSPA), 1,2-dioleoyl-3-trimethylammoniumpropane (DOTAP), and dioctylamidoglycyl-spermine (DOGS) [5]. These cationic lipidic systems are capable of interacting electrostatically with the negative charges of the phosphate groups of the nucleic acids, thus leading to their overall neutralization and the consequent compaction of the nucleotide fragment [5]. Analogously, antibacterial QASs act as ionic detergents, capable of rapidly binding the negatively charged bacterial wall and membranes by means of their cationic head [6]. Whereas, thanks to their long alkyl chain representing their hydrophobic (nonpolar) part, QASs can penetrate the nonpolar cell membrane, thus altering its permeability and leading to bacterial death [2]. Specifically, after absorption on the bacterial surface, by intercalating with the membranes and altering their normal structure, QASs succeed in modifying membrane permeability by pore formation, thus determining the loss of enzymes, coenzymes, and ions and compromising the biosynthetic activities of bacteria, triggering their death [3,4,6]. Interestingly, such mechanisms have little effect on the resistance and sensitivity of bacteria [7]. Unfortunately, QASs’ non-specific antimicrobial mechanism could translate into low selectivity for pathogens, significant toxicity toward eukaryotic cells, and hemolytic toxicity [3,8,9]. These drawbacks greatly hamper their administration in vivo by oral route and/or intravenous injection, limiting their use as surface disinfectants and antibacterial devices for epidermal treatment [3]. In this regard, designing new QASs or QAS composites with maintained antibacterial effects and reduced toxicity should be a new direction for the development of this field.

As recently reported, QASs could comprehend single-chain QASs, double-chain QASs, heterocyclic QASs [10], gemini QASs [11], and QASs with more than two positively charged moieties, depending on the number and structure of their long carbon chains and on the number and typology of their cationic heads [2,4]. Single-chain QASs are simply structured possess good antibacterial effects, and a wide range of antibacterial applications have been reported [12]. Anyway, the disadvantages of small-molecule single-chain QASs have been slowly revealed along with their use, such as their ease of producing resistance and their narrow killing spectrum [2]. On the contrary, double-chain QASs with more carbon chains are seen as more promising antimicrobial agents since they can combine the advantages of an excellent hydrophilic-oil balance and high antibacterial activities [13,14]. Additionally, the double-chain QAS known as didecyldimethylammonium bromide (DDAB) has demonstrated good removal capacity for biofilms formed by *S. aureus* and *P. aeruginosa* [15]. As for the number of cationic heads, multi-cationic QACs (multi-QACs) have been demonstrated to possess high activity against resistant bacterial species and to be less likely to induce antimicrobial resistance [16]. Multi-QACs with superior profiles against planktonic methicillin-resistant *Staphylococcus aureus* (MRSA) and biofilms have been reported by Jenning et al. [17]. The authors documented the ability of methicillin-sensitive *S. aureus* (MSSA) to develop resistance to several commercially available mono-QAC disinfectants and to an aryl bis-QAC, but no resistance to multi-QACs [17]. Moreover, the multi-QACs presented by Al-Khalifa and co-authors displayed significant advantages over their mono-cationic counterparts [18]. While similar red blood cell lysis was observed, superior activity against both Gram-negative bacteria and MRSA strains was reported [18]. Tetrakis quaternary ammonium compounds based on pentaerythritol having four cationic heads were synthesized by Vereshchagin et al., and evaluated against MRSA, *Escherichia coli*, and *Pseudomonas aeruginosa* [19]. The compound having a C8 chain was superior in its antibacterial activity to commercially available chlorhexidine, digluconate, and octenidine hydrochloride [19]. The same year, novel tris(4-alkylaminopyridin-1-ium) trichlorides with alkylcyanuric spacer were synthesized by the same authors and tested against five pathogenic bacterial strains, including *E. coli*, *K. pneumoniae*, *S. aureus*, *Acinobacter baumannii*, and *P. aeruginosa*, observing pronounced antibacterial properties [20]. More recently, Seferyan et al. reported that QASs with multiple heterocyclic cationic moieties prepared by an original isocyanuric acid alkylation method using available alkyl di-chlorides demonstrated higher levels of antibacterial activity against ESKAPE pathogens than widely used commercial QACs [16]. Such compounds possess high activity toward clinical bacterial strains and have also demonstrated a long-term biocidal effect without significant development of microorganism tolerance [16]. On this background, our first objective was to prepare a double-chain QAS with a quaternary ammonium head and two alkyl chains like those of the transfection reagents previously reported, hoping to obtain a compound with the same ability to bind negatively charged counterparts, including bacteria membranes. Anyway, it is known and reported that phosphonium salts (QPSs) with C8–C11 alkyl chains possess remarkable selective antibacterial effects, especially against Gram-positive species, thus attracting our attention [21,22]. Seddon and co-workers reported a series of tetra-alkyl phosphonium salts and studied their antimicrobial activity against several bacterial and fungal strains, observing an antibacterial activity depending on the length of the alkyl chains and the type of anion [22]. Interestingly, studies of comparison between QASs and QPSs showed that the latter possess antimicrobial activity higher than that of polymeric quaternary ammonium salts because of a difference in electronegativity between nitrogen and carbon atoms and phosphorous and carbon atoms [4,23]. Additionally, studies have shown that by replacing the cationic ammonium groups of QASs with phosphonium ones as in QPSs, phosphonium lipids with significantly lower cytotoxicity than the quaternary ammonium analogues can be developed [5].

Additionally, among QPSs, those based on triphenylphosphine have been studied to a greater extent, thus evidencing a broad spectrum of biological activity and antifouling properties [24]. On this new background, our scope was to synthesize a triphenyl alkyl phosphonium salt to be evaluated as a possible new antibacterial compound. In this regard, we thought it could be smart to obtain both the desired QPS and QAS compounds along the same synthetic path. Specifically, following a synthetic procedure experimented in the past for a doctoral thesis work [25], we prepared the cationic lipid 2,3-ditetradecyloxypropyl-1-trimethylammonium chloride (DTDTMA), here named **6** (Figure 1b), using the triphenyl phosphonium salt with a C11 alkyl chain, here named **1** (Figure 1a) as an intermediate for the Wittig reaction, necessary to prepare **6**, thus giving us the possibility of obtaining the desired QPS during the synthesis of the wanted QAS.

Compounds **1** and **6** were fully characterized by ATR-FTIR, NMR spectroscopy, and potentiometric titrations of their NH_3_^+^ groups. Titration was performed to have further confirmation of their structure by determining their experimental molecular weight (MW). Then, **1** and **6** were first screened against a selection of Gram-positive and Gram-negative MDR nosocomial isolates, evidencing significant antibacterial effects on Gram-positive isolates of *Enterococcus* and *Staphylococcus* genera. Secondly, their antibacterial effects were further studied on several additional strains of these species. Subsequently, their possible biocidal properties were investigated on opportunely selected strains of MRSA by performing time-killing experiments using **1**. Finally, to assess the clinical applicability of both compounds, dose-dependent cytotoxicity experiments were carried out on eukaryotic cells, using HepG2 human liver cell lines as an in vitro model.

## 2. Results and Discussion

### 2.1. Synthesis of Cationic Materials

Due to the reported antibacterial effects of both QASs [2,3,4,6,12,13,14] and QPSs [4,21,22,23], aiming at developing new antibacterial agents against MDR bacteria, the scope of this study was first to synthesize two molecules representative of both classes. We assumed to follow a single synthesizing path, which would have allowed us to achieve one type of compound (QPS **1**) during the synthesis of the other (QAS **6**). Moreover, we aimed at synthesizing a doble-chain QAS rather than a single-chain one because it was reported to be a better antibacterial agent than the latter [15]. As reported, the desired QAS compound should have possessed the ability to strongly interact with negatively charged materials, such as the negative constituents of bacterial membranes, thus causing irreversible damage and killing pathogens on contact. Since the commercially available DOTMA has already been reported to strongly interact with the negative charges of genetic materials [5], we were interested in preparing a QAS structurally like it. Particularly, QAS **6** should have contained two long alkenyl chains, a quaternary ammonium group and a chloride counterion, two ether bonds, and a central skeleton based on glycerol. Additionally, since liposomes containing lipids with a C14 alkyl chain and double bond in Δ11 were found to be very efficient in binding phosphate anions [26,27], we decided to equip our QAS with similar characteristics. On the other hand, concerning the desired QPS, we aimed at synthesizing an alkyl triphenilphosphonium derivative because it was reported to possess a broad spectrum of biological activities and antifouling properties [24]. Since the desired triphenyl alkyl phosphonium salts are the intermediate reagents to perform Wittig reactions to prepare alkenes, the Δ11 C14 vinylic chains of the designed QAS were prepared by just performing a Wittig reaction between the proper triphenyl alkyl phosphonium salt and the proper carbonylic compound. Specifically, following a synthetic procedure experimented in the past for a doctoral thesis work [25], we prepared the cationic lipid 2,3-ditetradecyloxypropyl-1-trimethylammonium chloride (DTDTMA) (**6**) (Figure 1a) according to Figure 1.

In the following Section 2.1.1, Section 2.1.2, Section 2.1.3, Section 2.1.4, Section 2.1.5 and Section 2.1.6, the synthesis and characterization of phosphonium salt **1** (QPS) and of all the intermediates prepared to achieve cationic lipide **6** (QAS) have been discussed in detail.

#### 2.1.1. (11-Hydroxyundecyl) Triphenyl Phosphonium Bromide (**1**)

11-bromo-undecan-1-ol, commercially available or synthesized by us (FTIR,^1^H, and ^13^C NMR spectra available in Appendix A in Appendix A), was reacted with triphenylphosphine (TPP), achieving the hydroxy-undecyl triphenyl phosphonium salt (**1**), as a syrup that crystallized as a white solid upon treatment under stirring with ethyl ether (Et_2_O) (Appendix A in Appendix A). It was characterized by FTIR (Figure 2), ^1^H NMR (Figure 3), ^13^C-NMR (Figure 4a), and DEPT-135 analyses (Figure 4b), which confirmed its structure and good degree of purity.

Particularly, in the FTIR spectrum, it was observable the strong and sharp band of the free OH stretching (3273 cm^−1^), the weak bands of the phenyls C-H stretching (3043 and 3015 cm^−1^), while the strong bands of the symmetric and asymmetric stretching of C–H of the several methylene groups of the alkyl chain were observable at 2923 and 2851 cm^−1^, respectively [28]. The C-H banding of the methylene groups was detected at 1463 cm^−1^, while the band around 1467 cm^−1^ could be due both to the stretching of aromatic C–C bonds and to the P-CH_2_ vibration [29]. The aromatic part of the prepared phosphonium salt was also confirmed by bands at 1482 and 1626 cm^−1^ (stretching vibrations of aromatic C=C bonds in the phenyl group) [28], and by the characteristic overtones in the range 1700–2000 cm^−1^.

The ^1^H NMR spectrum of **1** showed a series of broad multiplets in the range 1.18–1.61 ppm integrable for 18H and belonging to the C2–C10 (see the numbered structure of **1** in Figure 3) methylene groups. The broad singlet of the OH was detectable at 2.09 ppm, while at 3.60 ppm, it was found that the triplet belonged to the CH_2_OH group. The signal of the CH_2_-P^+^ group was instead detected as a broad multiplet at 3.70–3.80 ppm, whose multiplicity is due to the spin–spin coupling of protons with both the vicinal methylene group and with the phosphorus atom. Finally, a complex signal integrable for 15H was detected in the range 7.69–7.87 ppm, due to the proton atoms of the three phenyl rings.

At high fields (22.76–32.27 ppm) in the ^13^C-NMR spectrum of **1**, several signals due to the C1–C10 (see the numbered structure of **1** in Figure 4a) methylene groups of the aliphatic chain were detected, which appeared downwards-oriented in the DEPT-135 analysis (Figure 4b). Particularly, the signals in the range 22.76–23.42 ppm belonging to the carbon atoms C1 and C2 and the signal at 30.77, 30.98 ppm belonging to the C3 carbon atom appeared as doublets due to the spin–spin coupling with the phosphorous atom (*J*^1^*_CP_* = 33.01 Hz, *J*^2^*_CP_* = 20.9 Hz, and *J*^3^*_CP_* = 15.42 Hz). The signal of the methylene linked to the OH group was instead typically detected at 63.21 ppm. The signal of the quaternary C-1′ carbon atom of the benzene ring directly bonded to P^+^, which disappeared in the DEPT-135 analysis, was found at 118.43, 119.57 ppm, and showed a coupling constant of *J*^1^*_CP_* = 85.72 Hz. Whereas the *orto*- and *meta*-CH carbon atoms gave two doublets at 130.98, 131.14 ppm (CH*^o^* benzene ring, *J^o^* = 12.48 Hz) and at 134.14, 134.27 (CH*^m^* benzene ring, *J^m^* = 9.89 Hz). The *para*-CH was finally found at 135.54 ppm and showed an insignificant CP coupling.

#### 2.1.2. (*Z/E*)-11-Tetradecen-1-ol (**2**)

By a Wittig reaction using an excess of butyl lithium (BuLi), due to the presence of the hydroxyl group and propanoic aldehyde as a carbonylic compound, **1** was transformed into the desired C14, Δ11-alken-1-ol lipid chain (**2**), which was purified by column chromatography (Appendix A in Appendix A). The pure Δ11-tetradecen-1-ol (**2**) was obtained in the form of a colorless oil and as a mixture of *Z*/*E* isomers, with a significant predominance of *Z* chains, as evidenced by the ^1^H and ^13^C NMR spectra (Appendix A in Appendix A) and as reported in the literature [30]. Such assignment was further confirmed by comparing the NMR data of the predominant isomer with those described for (*Z*)-1-bromo-11-tetradecene [31]. Particularly, the signal of the methyl group appeared as a greater triplet centered at 0.95 ppm associated with a smaller one centered at 0.96 ppm, belonging to the *Z* and *E* isomers, respectively. A high broad peak at 1.28 ppm, integrable for 14H, and a multiplet centered at 1.56 ppm, integrable for 2H, were the signals of the C2–C9 methylene groups (Appendix A). The multiplet in the range 1.99–2.17 (4H) was given by the C10 and C13 methylene groups linked to the C11 and C12 carbon atoms of the double bond, while the OH and CH_2_OH groups provided the broad singlet at 1.91 ppm and the triplet centered at 3.62 ppm, respectively (Appendix A). Finally, the two hydrogen atoms of the double bond gave two multiplets, the more intense belonging to the *Z* isomer (5.27 ppm) and the smaller one to the molecules with *E* geometry (5.42 ppm) (Appendix A). By comparing the integrals of the signals relating to the CH_3_ protons in the ^1^H NMR spectrum (0.89–0.99 ppm) (Figure 5), it was possible to evaluate the *Z*/*E* ratio as equal to 82/18.

Although to a lesser extent, the presence of both *Z/E* isomers was also evidenced in the ^13^C NMR spectra, where we observed a split signal at 14.01 (*E*) and 14.39 (*Z*) ppm, upwards-oriented in the DEPT-135 experiment, belonging to the CH_3_ groups of the two isomers (Appendix A). A series of 10 peaks, of which 6 are very close to each other and are all downwards-oriented in the DEPT-135 analysis, belonged to ten of the eleven methylene groups of the alkyl chain. The CH_2_OH group provided a signal at 62.98 ppm, while the CH=CH double bond gave two split signals that remained upwards-oriented in the DEPT-135 experiment at 129.35 and 131.54 ppm, respectively. Concerning the FTIR spectrum (Appendix A in Appendix A), it was perfectly in accordance with that reported in literature [30] and available on the CAS Scifinder^n^ data base (https://scifinder-n.cas.org/searchDetail/substance/653a790b047a5b544a315636/substanceDetails, accessed on 20 January 2024). The purification of the mixture of alken-1-ol 2 also allowed the recovery of the unreacted phosphonium salt 1 (13%) usable for subsequent reactions. To avoid the use of large amounts of BuLi, in early experiments, compound 11-bromo-undecan-1-ol was protected with tetrahydropyran (THP) before reaction with TPP, resulting in a mixture of oily products that were difficult to purify and thus unsuitable to be employed successfully in the subsequent reaction.

#### 2.1.3. (*Z/E*)-11-tetradecenyl-1-Mesylate (**3**)

The esterification of **2** with commercially available mesyl chloride generated **3** as a mixture of isomers (77%) (Appendix A), maintaining the predominance of chains with *Z*-configuration, as confirmed by the ^1^H NMR spectrum (Appendix A). Briefly, as observed in the ^1^H NMR spectrum of **2** (Appendix A and Figure 5), the methyl group of **3** gave a greater triplet centered at 0.95 ppm (*Z* isomer) associated with a smaller one centered at 0.96 ppm (*E* isomer). The C2–C9 methylene groups of the alkyl chain provided a broad, high peak in the range 1.21–1.44 ppm integrable for 14H and a multiplet centered at 1.74 ppm integrable for 2H. The methylene groups linked to the C11 and C12 carbon atoms of the double bond (C10 and C13) gave a multiplet in the range 1.97–2.16 (4H) (Appendix A). The signal of the OH group of **2** previously detectable at 1.99 ppm, disappeared, while a singlet belonging to the CH_3_S group appeared. Additionally, the triplet of the CH_2_O group shifted at lower fields (4.22 ppm) with respect to that of the previous CH_2_OH (Appendix A). Finally, the two hydrogen atoms of the double bond gave two multiplets at 5.33 and 5.41 ppm belonging to the *Z* and *E* isomers, respectively (Appendix A). The 78/22 *Z*/*E* ratio was calculated as described in the previous section (Appendix A). FTIR and ^13^C NMR analyses completed the characterization of compound **3** (Appendix A in Appendix A). Particularly in the FTIR spectrum (Appendix A), it was observable a band at 3004 cm^−1^ (CH stretching of the double bonds), while the strong bands of the C–H symmetric and asymmetric stretching of the methyl and methylene groups were found at 2961, 2923, and 2855 cm^−1^. The band of CH=CH stretching was detected at 1652 cm^−1^, the band of methyl and methylene C-H banding was noticed at 1464 cm^−1^, and the typical bands of S=O stretching were observed at 1354 and 1175 cm^−1^. The presence of both *Z*/*E* isomers was also confirmed by the ^13^C NMR spectrum, where we observed two signals at 14.02 (*E* isomer) and 14.39 (*Z* isomer) ppm, upwards-oriented in the DEPT-135 analysis, belonging to the CH_3_ group of the two isomers (Appendix A). Ten peaks, which were downwards-oriented in the DEPT-135 analysis and belonged to ten of the eleven methylene groups of the alkyl chain, were found in the range of 20–33 ppm. The signal of the CH_3_S group appeared at 37.35 ppm, while the CH_2_O group provided a signal at 70.24 ppm. Finally, the CH=CH double bond gave two signals for each carbon atom, which remained upwards-oriented in the DEPT-135 experiment at 129.31 (*Z*) and 129.35 (*E*) ppm, as well as at 131.55 (*Z*) and 131.91 (*E*) ppm (Appendix A).

#### 2.1.4. *N*,*N*-Dimethyl-2,3-bis(tetradec-11-enyloxy)propylamine (**4**)

The mesylates obtained in the previous reaction were used to alkylate the freshly distilled *N*,*N*-dimethyl amino propane-2,3-diol after its deprotonation with NaH, thus obtaining the tertiary amine **4** (65%), which was purified on a chromatographic column (Appendix A in Appendix A). The column was necessary mainly to remove a mixture of substances of an ethereal nature deriving from side reactions between the mesylates. According to the possible combinations, compound **4** was obtained as a mixture of the *E*/*E*, *E*/*Z*, *Z*/*Z*, and *Z*/*E* molecules. Anyway, the *Z*/*E* ratio of the alkenyl chains was determined as in the previous Sections and was equal to 80/20 (Appendix A). Additionally, in Appendix A, the ^1^H and ^13^C NMR spectra, as well as the FTIR spectrum of **4,** have been reported. As observed in the ^1^H NMR spectra of **2** and **3**, the methyl groups of **4** gave a greater triplet centered at 0.95 ppm (*Z* chains) associated with a smaller one centered at 0.96 ppm (*E* chains). The C2–C9 methylene groups of the alkyl chains provided a high broad peak at 1.27 ppm integrable for 28H and a multiplet centered at 1.56 ppm integrable for 4H, while the methylene groups linked to the carbon atoms of the double bond (C10 and C13) gave a multiplet in the range 1.85–2.08 (8H) (Appendix A). The signal of the CH_3_S group of **3** disappeared, while a singlet belonging to the (CH_3_)_2_N group and a multiplet belonging to the CH_2_N group appeared at 2.37 and 2.51 ppm, respectively. A very complex signal given by both the CH_2_O, the RCH_2_O, and the CHO groups was observed in the range of 3.41–3.74 ppm. Finally, the 4 hydrogen atoms of the double bond gave two multiplets at 5.33 and 5.41 ppm belonging to the *Z* and *E* isomers, respectively (Appendix A). In the ^13^C NMR spectrum, we observed two signals at 14.02 (*E* isomer) and 14.40 (*Z* isomer) ppm, upwards-oriented in the DEPT-135 analyses, belonging to the CH_3_ group, and then several peaks that were downwards-oriented in the DEPT-135 analysis (20.52–32.59 ppm), belonging to 20 of the 24 methylene groups of the alkyl chains (Appendix A). The signal of the (CH_3_)_2_N group appeared at 45.90 ppm, while the CH_2_N group provided a signal at 60.82 ppm, which was downward-oriented in the DEPT-135. The RCH_2_O and CH_2_O groups gave signals at 71.68 and 70.13 ppm, respectively, while the signal of the CHO group, hidden in the ^13^C NMR spectrum, was well visible in the DEPT 135 experiment at 76.71 ppm (Appendix A). Finally, the CH=CH double bond gave two signals for each carbon atom, which remained upwards-oriented in the DEPT-135 experiment at 129.33 (*Z*) and 129.38 (*E*) ppm, as well as at 131.51 (*Z*) and 131.86 (*E*) ppm. In the FTIR spectrum (Appendix A), it was observable the band of the CH stretching of the double bonds at 3005 cm^−1^, the strong bands of the C-H symmetric and asymmetric stretching of the methyl and methylene groups (2961, 2925, and 2854 cm^−1^), the band of the C-H banding (1463 cm^−1^), while the typical bands of the S=O stretching disappeared.

#### 2.1.5. *N,N,N*-Trimethyl-2,3-bis(tetradec-11-enyloxy)propylammonium Iodide (**5**)

Treatment of **4** with excess iodomethane provided the quaternary ammonium compound **5** as iodide salt (Appendix A), having a mixture of *Z*/*E* chains in ratio 82/18 (Appendix A). Appendix A shows the FTIR spectrum of **5**, while Appendix A show their ^1^H and ^13^C NMR spectra, as well as a comparison between the ^13^C NMR spectrum of **5** and DEPT-135 analysis, respectively. Briefly, the FTIR spectrum showed the intense and sharp band of the double bond C-H stretching (3007 cm^−1^), and the C-H stretching bands of the methyl and methylene groups (2961, 2922, and 2852 cm^−1^), while less intense bands were observable at 1651, 1466, and 1117 cm^−1^ belonging to the C=C stretching, the C-H banding, and the C-N stretching, respectively (Appendix A). The signals detected in both the ^1^H and ^13^C NMR spectra of *Z*/*E*
**5** for the major compound *Z* were perfectly in agreement with those reported previously by Hurley et al. for the isolated *Z* isomer [32]. In particular, the methyl groups of **5** provided a greater triplet centered at 0.95 ppm (*Z* chains) associated with a smaller one centered at 0.96 ppm (*E* chains). The C2–C9 methylene groups of the alkyl chains gave a high broad peak at 1.27 ppm integrable for 28H and a multiplet centered at 1.56 ppm integrable for 4H, while the methylene groups linked to the C11 and C12 carbon atoms of the double bond (C10 and C13) gave a multiplet at 2.02 ppm (8H) (Appendix A). The signals at 2.37 and 2.51 ppm ((CH_3_)_2_NCH_2_ group) disappeared, while a high singlet belonging to the three methyl groups bonded to the quaternary nitrogen atom appeared at 3.50 ppm, while that of CH_2_N^+^ was found in the form of a multiplet at 4.00 ppm (Appendix A). A very complex signal given by both the CH_2_O, the RCH_2_O, and the CHO groups was observed in the range of 3.57–3.72 ppm. Finally, two multiplets at 5.31 and 5.41 ppm belonging to the CH=CH groups of *Z* and *E* chains, were detected (Appendix A). A magnification of the region of the spectrum in the range 0.8–3.65 ppm is available in Appendix A. In the ^13^C NMR spectra, we observed two signals at 14.02 (*E* isomer) and 14.40 (*Z* isomer) ppm, upwards-oriented also in the DEPT-135 analysis, belonging to the CH_3_ groups, and then several peaks that were downwards-oriented in the DEPT-135 analysis (20.52–32.58 ppm), belonging to 20 of the 24 methylene groups of the alkyl chains (Appendix A). The signal of the (CH_3_)_3_N^+^ group was found at 55.19 ppm, while that of the CH_2_N^+^ group was detected as a split signal at 68.07 (*E*) and 68.28 (*Z*) ppm, which was downward-oriented in the DEPT-135 analysis (Appendix A). The RCH_2_O and CH_2_O groups gave signals at 69.33 and 72.09 ppm respectively, while the signal of CHO group, was visible upwards-oriented both in the ^13^C NMR spectrum and in the DEPT-135 analysis at 73.57 ppm (Appendix A). Finally, the CH=CH double bond gave three signals that remained upwards-oriented in the DEPT-135 analysis at 129.33 (*Z+E*), 131.55 (*Z*), and 131.90 (*E*) ppm.

#### 2.1.6. *N,N,N*-Trimethyl-2,3-bis(tetradec-11-enyloxy)propylammonium Chloride (**6**)

*N*,*N*,*N*-trimethyl-2,3-bis(tetradec-11-enyloxy)propylammonium iodide (**5**) was subsequently treated with ion exchange resins (Appendix A) to obtain the corresponding chloride **6** as a yellow oil (*Z*/*E* = 82/18). The replacement of the counterion was necessary due to the lower cytotoxicity of the chloride ion [27]. Compound **6** differed from commercially available DOTMA for the length of the alkenyl chains (14 carbon atoms vs. 18) and the position of the double bond. Figure 6, Figure 7 and Figure 8 show the FTIR, ^1^H NMR, and ^13^C NMR spectra of **6**, while Appendix A show a comparison between the ^13^C NMR spectrum of **6** and its DEPT-135 analysis, between the ^1^H NMR spectra of **5** and **6**, and between their ^13^C NMR spectra and DEPT-135 analysis in the regions 66–76 ppm.

Briefly, the FTIR spectrum showed the sharp band of the double bond C-H stretching (3007 cm^−1^), and the C-H stretching bands of the methyl and methylene groups (2963, 2921, and 2853 cm^−1^), while less intense bands were observable at 1643, 1467, and 1123 cm^−1^, belonging to the C=C stretching, the C-H banding, and the C-N stretching, respectively (Figure 6).

In the ^1^H NMR spectrum of *Z*/*E*
**6**, the methyl groups provided a greater triplet centered at 0.95 ppm (*Z* chains) associated with a smaller one centered at 0.96 ppm (*E* chains). The C2-C9 methylene groups of the alkyl chains gave a high broad peak at 1.27 ppm integrable for 28H and a multiplet centered at 1.54 ppm integrable for 4H, while the methylene groups linked to the C11 and C12 carbon atoms of the double bond (C10 and C13) gave a multiplet at 2.02 ppm (8H) (Figure 7). The signal of the (CH_3_)_3_N^+^ group in the form of a high singlet was found at 3.49 ppm, while that of CH_2_N^+^ was found in the form of a multiplet at 4.06 ppm (Figure 7). A very complex signal given by both the CH_2_O, the RCH_2_O, and the CHO groups was observed in the range of 3.40–3.80 ppm. Finally, two multiplets at 5.34 and 5.41 ppm belonging to the CH=CH groups of *Z* and *E* chains were detected (Figure 7). Collectively, except for minimal changes in the chemical shift of some signals, the 1H NMR spectra of iodine and chloride salts were identical (Appendix A).

In the ^13^C NMR spectra, we observed two signals at 14.02 (*E* isomer) and 14.39 (*Z* isomer) ppm, upwards-oriented also in the DEPT-135 analysis, belonging to the CH_3_ groups, and then several peaks that were downwards-oriented in the DEPT-135 analysis (20.52–32.58 ppm), belonging to 20 of the 24 methylene groups of the alkyl chains (Figure 8 and Appendix A). The signal of the (CH_3_)_3_N^+^ group was found at 54.88 ppm, while that of the CH_2_N^+^ group was detected as a split signal at 67.98 (*E*) and 68.56 (*Z*) ppm, which was downward-oriented in the DEPT-135 analysis (Figure 8, Appendix A). As observable in Appendix A, in the spectrum of chloride salt **6**, the signals of *E* and *Z* chains were more distant than those of iodide salt. The RCH_2_O and CH_2_O groups gave signals at 69.36 and 72.07 ppm, respectively, while the signal of the CHO group was visible upwards-oriented both in the ^13^C NMR spectrum and in the DEPT-135 analysis at 73.71 ppm (Figure 8, Appendix A). Finally, the CH=CH double bond gave three signals, which remained upwards-oriented in the DEPT-135 analysis at 129.33 (*Z* + *E*), 131.56 (*Z*), and 131.91 (*E*) ppm. (Figure 8 and Appendix A).

### 2.2. Non-Aqueous Potentiometric Titration of **1** and **6**

Without recovering to more expensive GC/MS analyses, to have further confirmation of the molecular weight (MW) and structure of compounds **1** and **6**, we thought to titrate their ammonium groups, thus experimentally determining the NH_3_^+^ equivalents contained in an exactly weighted sample. Comparing the obtained results with those calculated according to the molecular weight required by the structures of **1** and **6**, we would have had confirmation of their mass and structure. In this regard, the emergence of non-aqueous titrations in the middle of the last century has enabled the possibility of determining both weak acids and bases not measurable in aqueous media [33,34,35]. Especially the titration of weak basic drugs with perchloric acid in glacial acetic acid medium is widely used. Titration in the acetic anhydride/acetic acid mixture enabled the direct non-aqueous titration of halide salts (mainly hydrochlorides) of organic bases and quaternary ammonium salts. Additionally, by adding mercury (II) acetate reagent to the quaternary ammonium salts solution, stable mercury (II) halide complexes and free acetate ions (equivalent to the base) that can be titrated with perchloric acid are formed [36,37]. On these considerations, we carried out the potentiometric titration of quaternary phosphonium and ammonium salts **1** and **6** in a mixture of anhydrous acetic acid (AcOH) and acetic anhydride (Ac_2_O) 30:70 (*v*/*v*) with 0.1 N HClO_4_, performing a slightly modified procedure previously described by us for the volumetric titration of ammonium salts [38,39,40]. Briefly, we adapted the protocol described by Pifer and Wollish, who applied this method for titrating quaternary ammonium salts [36], as described in the experimental Section. By plotting the measured mV values against the volumes of 0.1 N HClO_4_ solution added, we obtained the titration curves of **1** (Figure 9a) and **6** (Figure 9b) and the related first derivative (FD) curves.

The maxima of the FD represent the titration end points, which allowed us to find the volumes of titrating solution needed to titrate the ammonium groups of our samples and then their NH_3_^+^ equivalents. Table 1 reports the experimental details of titrations, the calculated NH_3_^+^ equivalents for **1** and **6** according to their molecular weight (MW), the experimentally determined NH_3_^+^ equivalents obtained by titrations, the experimental MW, the residuals, and the percentage error (%).

The experimental MWs were perfectly in agreement with the calculated ones, with a maximum error (%) of 0.14%, thus further confirming the structure of **1** and **6**.

### 2.3. Antibacterial Properties

#### 2.3.1. Antibacterial Activity of **1** and **6** by Determination of MIC Values (MICs)

For MICs evaluation, a total of 18 MDR strains of clinical origin were exploited. A first screening on a mini-selection of Gram-positive (1 *E. faecalis*, 1 *E. faecium*, 1 *S. aureus*, and 1 *S. epidermidis*) and Gram-negative species (1 *E. coli*, and 1 *Pseudomonas aeruginosa*) evidenced promising antibacterial effects against Gram-positive bacteria of the *Staphylococcus* and *Enterococcus* genus (Table 2). Based on these results, the antibacterial effects of **1** and **6** were further studied on additional strains of these genera, including 8 MDR Gram-positive enterococci (4 *E. faecalis* and 4 *E. faecium*) and 8 MDR Gram-positive staphylococci (4 *S. aureus* and 4 *S. epidermidis*). The results of this investigation have been reported in Table 3. In this study, as in other studies by us [41], 128 µg/mL was the threshold concentration over which a compound was considered inactive against a certain strain.

In Table 3, the MICs of the reference antibiotics commonly used to treat infections by enterococci and staphylococci were also included for comparison. Since the scope of our research was to find new antibacterial devices capable of counteracting bacteria resistance and promising for future clinical use, we have rationally chosen to compare the antibacterial effects of compounds **1** and **6** with those of currently available antibiotics commonly administered for counteracting infections sustained by the bacterial species considered, such as vancomycin and oxacillin. Since we did not aim at developing new disinfectants such as the commercially available benzalkonium chloride (BAC), cetylpyridinium chloride (CPC), or chlorhexidine (CHX) analogues of **6**, intended for daily use and not for clinical administration as therapeutics, we have thought nonsense about comparing the antibacterial effects of **1** and **6** with those of these compounds.

As evidenced in Table 2 and Table 3, all the clinical isolates used in this study were bacteria that had developed resistance to at least one or two antibiotics. Importantly, multidrug-resistant (MDR) *E. faecium* as well as *S. aureus* are included by WHO among the ESKAPE bacteria. This group of Gram-positive and Gram-negative bacteria is capable of evading or ‘escape’ commonly used antibiotics due to their increasing multi-drug resistance (MDR) [42]. As a result, they are the major cause of life-threatening opportunistic and hospital-acquired infections throughout the world [43]. In addition, along with the Gram-negative *P. aeruginosa*, *S. aureus* is one of the pathogens most often found in biofilms associated with medical devices, frequently used in healthcare facilities [44]. As observable in Table 2, while both **1** and **6** were inactive against *E. coli* and *P. aeruginosa* selected by us as representative of Gram-negative species, they showed very promising antibacterial effects against the tested strains of Gram-positive pathogens, including ESKAPE bacteria. As for QPS **1**, these results confirmed in part those recently reported by Shi et al., whose bis-phosphonium salts displayed very high MICs (≤64,000 µg/mL) against *E. coli* and *P. aeruginosa*, thus establishing their inactivity against Gram-negative species as in our case [45]. On the other hand, the MICs of compounds developed by these authors were remarkably higher (64–128 µg/mL) than those determined by us for **1** and **6** against MRSA, and according to our criteria, at the threshold of inactivity. In fact, the MICs we reported for **1** (4 μg/mL) were in general significantly lower, and even considering the best-performing compound developed by Shi et al. (4C) [45], **1** was 4 times more potent, thus suggesting that the antibacterial activity of QPS could not be associated with the number of cationic parts. Additionally, against MRSA, compound **1** displayed similar to higher antibacterial activity than five out of six multi-QACs developed by Vereshchagin et al. and displayed equal antibacterial activity to commercially available chlorhexidine digluconate [19]. As for QAS **6**, different results were found by Zhang et al. for six out of eight linear poly-isocyanide quaternary ammonium salts that showed from acceptable to good antibacterial activity against *E. coli* (27–105 µg/mL) [46]. However, against a *S. aureus* isolate for which no resistance was reported, such compounds demonstrate antibacterial activity similar to or extremely lower than that demonstrated by **6** against MRSA. Further experiments on an expanded population of MDR staphylococci and enterococci confirmed these very promising early observations. Particularly, **6**, reporting lower MIC values, especially against enterococci and especially against *E. faecium* (4 μg/mL), has shown promise as a new weapon against this worrying ESKAPE species. Conversely, **1** was more effective against staphylococci, with MICs not exceeding 8 μg/mL. In general, **1** was more potent than **6,** displaying the highest MIC of 16 µg/mL only against one out of four *E. faecium* and against three *E. faecalis*. Finally, the remarkable activity of **1** against ESKAPE pathogens such as *E. faecium* and MRSA must be considered of paramount importance and certainly underlines the relevance of the present study. In fact, MRSA is a particular isolate of *S. aureus*, distinct from other strains of the same species, against which not only the vast majority of β-lactam antibiotics no longer work but also several other drugs are inefficient due to the multiple traits of drug resistance acquired through horizontal gene transfer. [47].

Being very common in hospitals, prisons, and nursing homes (where immunocompromised patients and people with open wounds and/or invasive devices such as catheters are at greater risk of hospital-acquired infections), MRSA represents a global health threat and a clear ‘One Health’ problem. Moreover, MRA can spread and impact the environment, animals, and several human sectors [48].

Against MRSA, vancomycin is currently successful in approximately 49% of cases, and although the Infectious Disease Society of America recommends vancomycin and teicoplanin for the treatment of patients with MRSA pneumonia, using vancomycin can be complicated due to its inconvenient route of administration [49].

Unfortunately, several strains of MRSA have shown resistance even to vancomycin and teicoplanin since the late 1990s [50]. Oxazolidinones such as linezolid became available in the 1990s, but in 2001, cases of resistance towards such antibiotics were also reported [51].

Anyway, both for surgical site infections (SSIs) by MRSA [52] and for MRSA colonization in nonsurgical wounds such as traumatic wounds, burns, and chronic ulcers (i.e., the diabetic ulcer, pressure ulcer, arterial insufficiency ulcer, and venous ulcer), no conclusive evidence has been found about the best antibiotic regimen to treat the MRSA colonization [53]. In this alarming scenario, made up of missing epidemiologic evidence, a plethora of uncertainties due to the interindividual responses of patients to existing antibiotics, and the decreasing efficacy of available drugs, the development of new curative options against MRSA infections is urgent. Therefore, the overall merit of the present study lies in having identified, especially in Table 1**,** a potential new and potent antibacterial agent against MRSA.

#### 2.3.2. Time-Killing Curves

To assess whether the more potent of the two compounds developed here, i.e., **1**, was bactericidal or bacteriostatic, time-kill experiments were performed at concentrations equal to 4 × MIC on all MRSA, as reported in Table 2. Particularly, strains 17, 18, 187, and 195 were used. As depicted in Figure 10, the most representative curve obtained for strain 18, **1** showed bacteriostatic effects against MRSA, since a decrease of 1 log in the original cell number was evident after 6 h of exposure. During the next 24 h, a slightly additional decrease was observed, and no regrowth occurred.

### 2.4. Cytotoxicity Experiments on Human Liver Cells

In addition to solubility in water, a sufficiently high value of the selectivity index (SI) is an essential requirement to render a new molecule worthy of consideration for further studies and future development as a new therapeutic agent. In microbiology, to have an acceptable value of SI, a new antimicrobial agent should have low MICs on bacteria, associated with a high concentration able to induce 50% of cell death (expressed as IC_50_) on eukaryotic cells. In this regard, the SI value is given by the Equation (1) and provides a measure of the selectivity of the antimicrobial agent for bacteria.
SI = IC_50_/MIC(1)

To obtain the SI values of QPS **1** and QAS **6,** we carried out dose-dependent cytotoxicity experiments, by MTT assay, on human liver cells (HepG2) [54,55]. The results were used to compute the IC_50_, and the obtained IC_50_ and the MIC values reported in Table 2 were then used to calculate the SI values of **1** and **6** against each isolate considered in this study, as discussed in the following section.

#### Concentration-Dependent Cytotoxicity Test

In view of a potential human application of **1** and **6**, we tested their possible toxic effects on human HepG2 liver cell lines. The cells were exposed for 24 h to different concentrations of both compounds (2–160 µg/mL and 8–640 µg/mL for **1** and **6,** respectively), based on the observed lower and higher MICs. The values of cell viability were expressed as mean ± standard deviation (SD) and were plotted vs. concentrations, obtaining both bar graphs (Figure 11 and Figure 12) and dispersion graphs (Appendix A).

According to the results shown in Figure 11 and Figure 12, compounds **1** and **6** demonstrated a similar concentration-dependent cytotoxicity pattern towards HePG2 cell lines, with a non-significant induced-cell death for both QPS **1** and QAS **6**, when used at the lower MIC of 2 µg/mL and 8 µg/mL.

For a more rigorous evaluation, we calculated their IC_50_ using GraphPad Prism software 10 and fitted the data with non-linear regressions. Particularly, we plotted the Log10 of the concentrations vs. the normalized values of cell viability observed (Appendix A). The calculated IC_50_ was 27.95 ± 13.33 µg/mL (QPS **1**) and 30.80 ± 21.28 µg/mL (QAS **6**), respectively, thus confirming similar cytotoxicity, with QPS **1** slightly more cytotoxic than QAS **6**. The SI of both compounds was calculated using Equation (1), and the MICs reported in Table 3 reveal that the compound is more suitable for further studies and future development for clinical use. The results have been included in the following Table 4.

Concerning the SI values necessary for establishing the suitability of a new molecule for further studies and clinical development, the reported opinions are conflicting. Few authors stated that SI values ≥ 10 are necessary to make a molecule worthy of further investigation [56,57], while Weerapreeyakul et al. [58] proposed SI values ≥ 3 to define a clinically applicable molecule as an anticancer agent. In microbiology, Adamu et al. [59,60] reported SI values ≤ 5.2 for South African plant leaf extracts with antibacterial properties. Famuyide et al. [61] stated that antibacterial plant extracts could be considered bioactive and non-toxic if SI >1, while Nogueira and Estolio do Rosario reported that the SI should not be less than 2 [62]. In this scenario of contrasting opinions, while QAS (**6**) could be developed as a therapeutic to specifically treat infections supported by *E. faecium*, the SI of QPS (**1**) was sufficiently high against all MDR strains considered.

## 3. Materials and Methods

### 3.1. Chemicals and Instruments

All reagents and solvents were from Merck (formerly Sigma-Aldrich, Darmstadt, Germany) and were purified by standard procedures. The organic solutions were dried over anhydrous magnesium sulphate and evaporated using a rotatory evaporator operating at a reduced pressure of about 10–20 mmHg. The melting ranges of the solid compounds in this study were determined on a 360 D melting point device with a resolution of 0.1 °C (MICROTECH S.R.L., Pozzuoli, Naples, Italy). The melting points and boiling points are uncorrected. Attenuated total reflectance (ATR), Fourier transform infrared (FTIR), ^1^H and ^13^C NMR analyses were carried out on the same instruments as previously reported [41]. Column chromatography was performed on Merck silica gel (70–230 mesh). Potentiometric titrations were carried out using a Hanna Micro-processor Bench pH Meter (Hanna Instruments Italia srl, Ronchi di Villafranca Padovana, Padova, Italy), which was calibrated using standard solutions at pH = 4, 7, and 10 before titrations. Thin layer chromatography (TLC) was carried out using aluminum-backed silica gel plates (Merck DC-Alufolien Kieselgel 60 F254, Merck, Washington, DC, USA), and detection of spots was made by UV light (254 nm) using a Handheld UV Lamp, LW/SW, 6W, UVGL-58 (Science Company^®^, Lakewood, CO, USA).

### 3.2. Preparation of a Butyl Lithium (BuLi) Solution in Et_2_O

In a three-necked flask equipped with a mechanical stirrer, dripping funnel, thermometer, and nitrogen valve, lithium (Li) (4.30 g, 0.62 mol) was suspended in anhydrous Et_2_O (100 mL). A solution of 1-bromobutane (34.50 g, 0.25 mol, 27 mL) in anhydrous Et_2_O (50 mL) was prepared in the dripping funnel. 1-bromobutane solution (2 mL) was added drop by drop to the suspension and cooled to −30 °C, noting a slight increase in temperature, a progressive clouding, and the formation of white patches on the surface of the metal, indicative of the reaction starting. The dripping of the 1-bromobutane solution is completed in approximately 30 min, maintaining the temperature at −10 °C. The suspension was left under stirring for an hour, maintaining the temperature below 10 °C, then decanted into a 100 mL test tube and left to settle in the fridge overnight. Aliquots of the clear solution (1 mL) were withdrawn, which were diluted with 5 mL of anhydrous THF, added with a few drops of a phenolphthalein solution (0.1% in EtOH) as an indicator, added with 0.1 N HCl until the pink color disappeared (about 10 mL), and titrated with 0.1 N NaOH. Titration was carried out in triplicate, and the results were expressed as mean ± standard deviation (SD).

### 3.3. (11-Hydroxyundecyl) tri-Phenyl Phosphonium Bromide (**1**)

11-bromoundecan-1-ol (**1**) (9.80 g, 0.04 mol) and triphenylphosphine (TPP) (12.28 g, 0.05 mol) were dissolved in CH_3_CN (162 mL). The reaction mixture was kept under stirring in a nitrogen stream at reflux for 24 h, after which the solvent was removed under reduced pressure, obtaining a yellowish oil that was taken up with anhydrous Et_2_O (50 mL) under vigorous mechanical stirring to promote coagulation. After approximately 15 min, **1** was obtained in the form of a white solid, which was filtered and brought to a constant weight at reduced pressure (19.95 g, 0.04 mol, 99%). M. p. = 90–92 °C. FTIR (KBr, cm^−1^): 3407, 3279 (OH); 2924, 2853 (CH_2_); 2000–1700 (overtones); 1626, 1482 (C=C stretching); 1467 (C=C and/or P^+^-CH_2_ stretching); 1463 (aliphatic C-H banding). ^1^H-NMR (CDCl_3_, 300 MHz, δ ppm): 1.18–1.61 (m, 18H); 2.09 (broad s, 1H, OH); 3.60 (t, 2H, *J* = 6.6 Hz); 3.70–3.80 (m, 2H, CH_2_P^+^); 7.69–7.87 (m, 15H). ^13^C-NMR (CDCl_3_, 75.5 MHz, δ ppm): 22.76, 23.15, 23.20, 23.42 (dd, CH_2_(1) and CH_2_(2), *J^1^_CP_* = 33.01 Hz, *J^2^_CP_* = 20.9 Hz); 26.22 (CH_2_), 29.56–29.83 (5CH_2_), 30.77,30.98 (d, CH_2_(3), *J^3^_CP_* = 15.42 Hz), 33.27 (CH_2_), 63.21 (CH_2_OH); 118.43, 119.57 (C-1 quaternary benzene ring bonded to P^+^, *J^1^_CP_* = 85.72 Hz); 130.98, 131.14 (CH*^o^* benzene ring, *J^o^* = 12.48 Hz); 134.14, 134.27 (CH*^m^* benzene ring, *J^m^* = 9.89 Hz); 135.54 (C-4 benzene ring).

### 3.4. (Z/E)-11-Tetradecen-1-ol (**2**)

In a 250 mL three-neck flask, equipped with a mechanical stirrer, condenser, drip funnel, and nitrogen valve, **1** (19.95 g, 0.04 mol) was suspended in anhydrous (tetrahydrofuran) THF (108 mL), cooled to −40 °C, and added drop by drop with a solution of 0.856 M BuLi in Et_2_O (82.5 mL), noticing a color change first to yellow, then to orange, and finally to intense red. The reaction mixture was kept under stirring in a stream of nitrogen at −20 °C for 2 h, then treated with a solution of propionic aldehyde (2.66 g, 0.05 mol, 3.3 mL) in THF (7.2 mL), noting the solution color change from red to yellow. The reaction mixture was kept under stirring in a stream of nitrogen at −20 °C for another 2 h, hydrolyzed with sat NH_4_Cl (15 mL), and kept stirring for an additional 30 min. The precipitated solid was filtered off and washed with THF. The organic phase and washing solvent were washed with H_2_O (15 mL) and sat. NaCl. (15 mL) and dried over Na_2_SO_4_ overnight. Removal of the solvent at reduced pressure provided crude **2,** which was purified by a silica gel chromatography column using a 40–60 °C petroleum ether/ethyl acetate (EtOAc) = 1/9 mixture as eluent. Colorless oil (4.61 g, 0.02 mol, 50%). FTIR (KBr disc, cm^−1^): 3337 (OH); 3005 (CH=); 2961, 2926, 2854 (CH_3_, CH_2_); 1652 (C=C); 1463 (C-H banding). ^1^H-NMR (CDCl_3_, 300 MHz, δ ppm): 0.95 (t, 3H, *J* = 7.5 Hz, CH_3_, *Z*); 0.95 (t, 3H, *J* = 7.5 Hz, CH_3_, *E*); 1.27–1.40 (m, 14H, CH_2_, *Z*+*E*); 1.53–1.58 (m, 2H, CH_2_, *Z*+*E*); 1.91 (br s, 1H, *Z*+*E*); 1.98–2.17 (m, 4H, CH_2_, *Z*+*E*); 3.62 (t, 2H, *J* = 6.6 Hz, CH_2_OH, *Z*+*E*); 5.27 (m, 2H, CH=CH, *Z*); 5.42 (m, 2H, CH=CH, *E*). ^13^C-NMR (CDCl_3_, 75.5 MHz, δ ppm): 14.02 (CH_3_, *E*); 14.41 (CH_3_, *Z*); 20.53 (CH_2_, *Z*+*E*); 25.63 (CH_2_, *E*); 25.79 (CH_2_, *Z*); 27.13 (CH_2_, *Z*+*E*); 29.31 (CH_2_, *Z*+*E*); 29.48 (CH_2_, *Z*+*E*); 29.57 (CH_2_, *Z*+*E*); 29.61 (CH_2_, *Z*+*E*); 29.65 (CH_2_, *Z*+*E*); 29.81 (CH_2_, *Z*+*E*); 32.81 (CH_2_, *Z*+*E*); 62.98 (CH_2_OH, *Z*+*E*); 129.35 (CH=, *Z*); 129.39 (CH=, *E*); 131.54 (CH=, *Z*); 131.89 (CH=, *E*).

### 3.5. (Z/E)-11-Tetradecenyl-1-mesylate (**3**)

**2** (4.40 g, 0.02 mol) dissolved in CH_2_Cl_2_ (5 mL) was added with Et_3_N (4.44 g, 0.04 mol, 6 mL). The solution was cooled to 0 °C and added drop by drop with a methyl sulphonyl chloride solution (3.37 g, 0.23 mol, 2 mL) in dichloromethane (CH_2_Cl_2_) (5 mL), noting a progressive clouding of the solution. The mixture was left to reach room temperature and was kept under stirring in a stream of nitrogen for 90 min (TLC petroleum ether 40–60 °C/EtOAc = 9/1, using an alcoholic solution of phosphomolybdic acid as a highlighter). After the addition of CH_2_Cl_2_ to bring the volume to 100 mL, the mixture was washed with H_2_O (30 mL), HCl 1/1 (30 mL), sat. NaHCO_3_. (30 mL) and sat. NaCl (30 mL). The organic phase was separated and dried over anhydrous Na_2_SO_4_ overnight. Elimination of the solvent at reduced pressure provided crude **3,** which was purified by column chromatography on silica gel using a mixture of petroleum ether 40–60 °C/CH_2_Cl_2_ = 1/4 as eluent. Colorless oil (4.61 g, 0.02 mol, 77%). FTIR (KBr disc, cm^−1^): 3004 (CH=); 2961, 2926, 2855 (CH_3_, CH_2_); 1652 (C=C), 1464 (C-H banding); 1354 (S=O); 1176 (S=O and C-O). ^1^H-NMR (CDCl_3_, 300 MHz, δ ppm): 0.95 (t, 3H, *J* = 7.5 Hz, CH_3_, *Z*); 0.96 (t, 3H, *J* = 7.5 Hz, CH_3_, *E*); 1.28–1.44 (m, 14H, CH_2_, *Z*+*E*); 1.72–1.79 (m, 2H, CH_2_, *Z*+*E*); 1.90–2.16 (m, 4H, CH_2_, *Z*+*E*); 3.00 (s, 3H, CH_3_S, *Z*); 3.001 (s, 3H, CH_3_S, *E*); 4.22 (t, 2H, *J* = 6.6 Hz, CH_2_O, *Z*+*E*); 5.33 (m, 2H, CH=CH, *Z*); 5.41 (m, 2H, CH=CH, *E*). ^13^C-NMR (CDCl_3_, 75.5 MHz, δ ppm): 14.01 (CH_3_, *E*); 14.39 (CH_3_, *Z*); 20.52 (CH_2_, *Z*+*E*); 25.44 (CH_2_, *Z*); 25.61 (CH_2_, *E*); 27.09 (CH_2_, *Z*+*E*); 29.04 (CH_2_, *Z*+*E*); 29.16 (CH_2_, *Z*+*E*); 29.26 (CH_2_, *Z*+*E*); 29.42 (CH_2_, *Z*+*E*); 29.49 (CH_2_, *Z*); 29.66 (CH_2_, *E*); 29.77 (CH_2_, *Z*+*E*); 32.56 (CH_2_, *Z*+*E*); 37.36 (CH_3_S, *Z*+*E*); 70.23 (CH_2_O, *Z*+*E*); 129.31 (CH=, *Z*); 129.35 (CH=, *E*); 131.55 (CH=, *Z*); 131.91 (CH=, *E*).

### 3.6. N,N-Dimethyl-2,3-bis(tetradec-11-enyloxy)propylamine (**4**)

In a 250 mL two-neck flask equipped with a magnetic stirrer, condenser, and nitrogen valve, NaH was introduced as a dispersion in 60% mineral oil (357.4 mg, 15.5 mmol), which was washed abundantly with anhydrous pentane. The gray solid obtained was suspended in toluene (31 mL) and added to *N*,*N*-dimethylamino-propane-1,2-diol (617.0 mg, 5.2 mmol, 420 μL), observing the immediate formation of a pink precipitate. The suspension was kept under stirring in a stream of nitrogen at 50 °C for 1 h, then added with **3** (4.51 g, 15.5 mmol) and left under stirring in a stream of nitrogen at reflux for 72 h. After cooling to 0 °C, the suspension was hydrolyzed with H_2_O (100 mL), observing the formation of a white precipitate, and extracted with EtOAc (3 × 50 mL). The organic phase was washed with sat. NaHCO_3_ (30 mL) and sat. NaCl (30 mL), while the aqueous washes were re-extracted with EtOAc (50 mL), and the combined organic phases were dried over anhydrous Na_2_SO_4_ overnight. Removal of the solvent at reduced pressure provided crude **4,** which was purified by a silica gel chromatography column using the mixture CH_2_Cl_2_/MeOH = 9/1 as eluent. Yellowish oil (1.72 g, 3.4 mmol, 65%). FTIR (KBr disc, cm^−1^): 3005 (CH=); 2961, 2925, 2854 (CH_3_, CH_2_); 1463 (C-H banding), 1118 (C-N stretching). ^1^H-NMR (CDCl_3_, 300 MHz, δ ppm): 0.95 (t, 6H, *J* = 7.5 Hz, 2CH_3_, *Z*); 0.96 (t, 6H, *J* = 7.5 Hz, 2CH_3_, *E*);1.27 (m, 28H, 14 CH_2_, *Z*+*E*); 1.56 (m, 4H, 2CH_2_, *Z*+*E*); 1.97–2.17 (m, 8H, 4CH_2_, *Z*+*E*); 2.37 (s, 6H, (CH_3_)_2_N, *Z*+*E*); 2.45–2.59 (m, 2H, CH_2_N, *Z*+*E*); 3.41–3.64 (m, 7H, CH_2_O+CHO, *Z*+*E*); 5.33 (m, 4H, 2 CH=CH, *Z*); 5.41 (m, 4H, 2 CH=CH, *E*). ^13^C-NMR (CDCl_3_, 75.5 MHz, δ ppm): 14.02 (CH_3_, *Z*); 14.39 (CH_3_, *E*); 20.52 (CH_2_, *Z*+*E*); 25.61–27.17 (4CH_2_, *Z*+*E*); 29.17–32.88 (7CH_2_, *Z*+*E*); 45.90 ((CH_3_)_2_N, *Z*+*E*); 60.82 (CH_2_N, *Z*+*E*); 70.13 (CH_2_O, *Z*+*E*); 71.68 (2 CH_2_O, *Z*+*E*); 77.51 (CHO, *Z*+*E*); 129.34 (CH=, *Z*); 129.38 (CH=, *E*); 131.51 (CH=, *Z*); 131.86 (CH=, *E*).

### 3.7. N,N,N-Trimethyl-2,3-bis(tetradec-11-enyloxy)propylammonium Iodide (**5**)

Compound **4** (1.59 g, 3.1 mmol) and CH_3_I (4.51 g, 31.8 mmol, 3 mL) were introduced into a 25 mL single-neck flask, equipped with a magnetic stirrer, and left under stirring at room temperature for 18 h after wrapping the ball in a black cloth. Then, the excess CH_3_I was eliminated under reduced pressure, obtaining a dark brown oil, which was purified by a silica gel chromatography column using the chloroform (CHCl_3_)/methanol (MeOH) = 4/1 mixture as eluent. Compound **5** was obtained as a yellow solid (1.05 g, 1.6 mmol, 52%). FTIR (KBr disc, cm^−1^): 3007 (CH=); 2962, 2922, 2852 (CH_3_, CH_2_); 1651 (C=C); 1466 (C-H banding); 1117 (C-N stretching); 1175 (C-O). ^1^H-NMR (CDCl_3_, 300 MHz, δ ppm): 0.95 (t, 6H, *J* = 7.5 Hz, 2CH_3_, *Z*); 0.96 (t, 6H, *J* = 7.5 Hz, 2CH_3_, *E*); 1.27 (m, 28H, 14CH_2_, *Z*+*E*); 1.56 (m, 4H, 2CH_2_, *Z*+*E*); 1.99–2.17 (m, 8H, 4CH_2_, *Z*+*E*); 3.41–3.72 (m, 7H, CH_2_O+CHO, *Z*+*E*); 3.50 (s, 9H, (CH_3_)_3_N^+^, *Z*+*E*); 3.98–4.03 (m, 2H, (CH_2_N^+^, *Z*+*E*); 5.34 (m, 4H, CH=, *Z*); 5.41 (m, 4H, CH=, *E*). ^13^C-NMR (CDCl_3_, 75.5 MHz, δ ppm): 14.02 (CH_3_, *Z*); 14.39 (CH_3_, *E*); 20.52 (CH_2_, *Z*+*E*); 25.44–27.16 (4CH_2_, *Z*+*E*); 29.08–32.58 (7CH_2_, *Z*+*E*); 55.19 (CH_3_)_3_N^+^, *Z*+*E*); 68.07 (CH_2_N, *E*); 68.28 (CH_2_N, *Z*); 69.33 (CH_2_O, *Z*+*E*); 72.09 (2RCH_2_O, *Z*+*E*); 73.57 (CHO, *Z*+*E*); 129.33 (CH=, *Z*+*E*); 131.55 (CH=, *Z*); 131.89 (CH=, *E*).

### 3.8. N,N,N-Trimethyl-2,3-bis(tetradec-11-enyloxy)propylammonium Chloride (**6**)

In a 100 mL single-neck flask equipped with a magnetic stirrer, **5** (1.02 g, 1.6 mmol) was dissolved in CH_2_Cl_2_/MeOH = 1/1 (78 mL) and added with Amberlite-IRA-400(Cl) resins (4 g). The reaction mixture was kept under stirring at room temperature for one night, then the resins were separated by filtration and washed with the same mixture of solvents. Evaporation of the solvents at reduced pressure provided **6** in the form of a yellow oil with a tendency to solidify (876.6 mg, 1.6 mmol; 99%). FTIR (KBr disc, cm^−1^): 3007 (CH=); 2963, 2924, 2853 (CH_3_, CH_2_); 1643 (C=C); 1467 (C-H banding); 1176 (C-O); 1123 (C-N stretching). ^1^H-NMR (CDCl_3_, 300 MHz, δ ppm): 0.95 (t, 6H, *J* = 7.5 Hz, 2CH_3_, *Z*); 0.96 (t, 6H, *J* = 7.5 Hz, 2CH_3_, *E*); 1.27 (m, 28H, 14CH_2_, *Z*+*E*); 1.54 (m, 4H, 2CH_2_, *Z*+*E*); 1.98–2.08 (m, 8H, 4CH_2_, *Z*+*E*); 3.43–3.80 (m, 7H, CH_2_O+CHO, *Z*+*E*); 3.49 (s, 9H, (CH_3_)_3_N^+^, *Z*+*E*); 3.90–4.10 (m, 2H, (CH_2_N^+^, *Z*+*E*); 5.34 (m, 4H, CH=, *Z*); 5.41 (m, 4H, CH=, *E*). ^13^C-NMR (CDCl_3_, 75.5 MHz, δ ppm): 14.02 (CH_3_, *Z*); 14.39 (CH_3_, *E*); 20.52 (CH_2_, *Z*+*E*); 25.61–27.12 (4CH_2_, *Z*+*E*); 29.09–32.58 (7CH_2_, *Z*+*E*); 54.88 (CH_3_)_3_N^+^, *Z*+*E*); 67.98 (CH_2_N, *E*); 68.60 (CH_2_N, *Z*); 69.36 (CH_2_O, *Z*+*E*); 72.07 (2RCH_2_O, *Z*+*E*); 73.71 (CHO, *Z*+*E*); 129.33 (CH=, *Z*+*E*); 131.56 (CH=, *Z*); 131.91 (CH=, *E*).

### 3.9. Potentiometric Titrations of **1** and **6**

The potentiometric titration of quaternary phosphonium and ammonium salts **1** and **6** was carried out in non-aqueous medium (mixture of anhydrous acetic acid (AcOH) and acetic anhydride (Ac_2_O) 30:70 (*v*/*v*)) with HClO_4_, performing a slightly modified procedure previously described by us for the volumetric titration of ammonium salts [38,39,40]. A similar protocol was in fact described by Pifer and Wollish, who applied this method for titrating quaternary ammonium salts [36]. Briefly, exacted weighted samples of **1** (280.0 mg) or **6** (245.5 mg) were dissolved in AcOH/Ac_2_O 30/70 (5 mL), treated with 2–4 mL of a solution of mercury acetate (1.5 g) in AcOH (25 mL), and titrated with a standardized 0.1 N solution of HClO_4_ in AcOH/Ac_2_O, prepared as described in the following section, using potentiometric endpoint detection. The titrations were performed under efficient stirring with a magnetic stirrer, at room temperature (25 ± 2 °C). Millivolts were measured every 0.5 mL up to 3–4 mL and every 0.1 mL in the vicinity of the calculated end point up to the addition of 6 mL (**1**) and 5.4 mL (**6**) of 0.1 N HClO_4_. Titrations were made in triplicate, and the measurements were reported as mean ± SD.

#### Preparation of a 0.1 M Perchloric Acid Volumetric Solution

The 0.1 M perchloric acid volumetric solution was prepared by diluting 8.5 mL of 70–73 wt% perchloric acid with 900 mL of anhydrous acetic acid and 30 mL of acetic anhydride and then diluting to 1000 mL with anhydrous acetic acid. Perchloric acid was standardized by titration against potassium hydrogen phthalate.

### 3.10. Microbiology

#### 3.10.1. Microorganisms

A total of 20 isolates, belonging to a collection of MDR Gram-positive and Gram-negative species from the University of Genova, were used in this study. All were clinical strains isolated from human specimens and identified by VITEK^®^ 2 (Biomerieux, Firenze, Italy) or matrix-assisted laser desorption/ionization time-of-flight (MALDI-TOF) mass spectrometric technique (Biomerieux, Firenze, Italy). The 18 MDR isolates include, for a first screening, 4 Gram-positive species (1 *E. faecalis*, 1 *E. faecium*, 1 *S. aureus*, and 1 *S. epidermidis*) and 2 Gram-negative species (1 *E. coli* and 1 *Pseudomonas aeruginosa*). For further investigations, additional strains of Gram-positive species were used, including 8 MDR Gram-positive enterococci (4 *E. faecalis* and 4 *E. faecium*) and 8 MDR Gram-positive staphylococci (4 *S. aureus* and 4 *S. epidermidis*).

#### 3.10.2. Determination of the MICs

To investigate the antimicrobial activity of **1** and **6** on the described pathogens, their Minimal Inhibitory Concentrations (MICs) were determined by following the microdilution procedures detailed by the European Committee on Antimicrobial Susceptibility Testing EUCAST [63].

Briefly, after overnight incubation, cultures of bacteria were diluted to yield a standardized inoculum of 1.5 × 10^8^ CFU/mL. Appropriate aliquots of each suspension were added to 96-well microplates containing the same volumes of serial 2-fold dilutions (ranging from 1 to 128 μg/mL) of **1** or **6** to yield a final concentration of about 5 × 10^5^ cells/mL. The plates were then incubated at 37 °C. After 24 h of incubation at 37 °C, the lowest concentration of **1** or **6** that prevented visible growth was recorded as the MIC. All MICs were obtained at least in triplicate, and the mode value was selected as the reported MIC.

#### 3.10.3. Killing Curves

Killing curve assays for **1** were performed on three representative isolates of *S. aureus* (strains 17, 18, 187, and 195, all MRSA), as previously reported [64]. Experiments were performed over 24 h at **1** concentration four times the MIC for all strains.

A mid-logarithmic phase culture was diluted in Mueller–Hinton (MH) broth (Merck, Darmstadt, Germany) (10 mL) containing 4 × MIC of the selected compound in order to give a final inoculum of 1.0 × 10^5^ CFU/mL. The same inoculum was added to cation-supplemented Mueller–Hinton broth (CSMHB) (Merck, Darmstadt, Germany) as a growth control. Tubes were incubated at 37 °C with constant shaking for 24 h. Samples of 0.20 mL from each tube were removed at 0, 30 min, 2, 4, 6, and 24 h, diluted appropriately with a 0.9% sodium chloride solution to avoid carryover of **1** or **6** being tested, plated onto MH plates, and incubated for 24 h at 37 °C. Growth controls were run in parallel. The percentage of surviving bacterial cells was determined for each sampling time by comparing colony counts with those of standard dilutions of the growth control. Results have been expressed as log10 of viable cell numbers (CFU/mL) of surviving bacterial cells over a 24 h period. The bactericidal effect was defined as a 3 log10 decrease in CFU/mL (99.9% killing) of the initial inoculum. All time-kill curve experiments were performed in triplicate.

### 3.11. MTT Cell Viability Assay

To assess the toxicological properties of 1 and 6, the MTT (3-(4,5-dimethylthiazol-2-yl)-2,5-diphenyltetrazolium bromide) cell viability assay was performed. Briefly, HepG2 cells were plated at 20,000 cells/well into 96-well plates. The cells were maintained in complete Dulbecco’s Modified Eagle Medium (DMEM; Euroclone, Cat# ECM0728L) containing 10% Fetal Bovine Serum (Euroclone, Cat# ECS0180L), 1% glutamine (Euro-clone, Cat# ECB3004D), and 1% Penicillin/Streptomycin (Euroclone, Cat# ECB3001D) at 37 °C in a 5% CO_2_ atmosphere for 24 h. The DMEM was replaced with fresh media containing, respectively, compound 1 at the concentration range of 2–160 µg ml^−1^, compound 6 at the concentration range of 8–640 µg ml^−1^, DMSO at the concentration range of 2–0.01 µL ml^−1^, and DMEM w/o DMSO as the negative control condition (CTRL). A cytotoxic curve with an increasing concentration of DMSO (1–100 µL mL^−1^) has been performed as a positive control for the MTT assay. The cells were incubated at 37 °C in 5% CO_2_ for another 24 h. After that, the media was removed, and cells were washed with PBS. Aliquots (200 µL) of serum-free medium containing MTT (Merck, Cat #M5655; 0.25 mg/mL MTT) were added to each well and incubated at 37 °C for 2 h. After removing the medium, 200 µL of DMSO solution (Merck, Cat #276855; 0.25 mg/mL MTT) were added to each well and horizontally shaken for 10 min to allow DMSO to solubilize the formed formazan crystals, making a homogenous solution. The 570 nm wavelength light absorption was then measured spectrophometrically in each well and converted into the OD (optical density) unit using the Spectra Max 340 PC (Molecular Devices, San Jose, CA, US). The cell survival ratio expressed as cell viability percentage (%) was evaluated based on the treated group results (compounds **1** and **6**) vs. the untreated group results (DMEM w/o DMSO, CTLR) and was calculated as follows: cell viability (%) = (OD treated cells − OD blank)/(OD untreated cells − OD blank) × 100%. Graphs and statistics were generated by GraphPad Prism (Version 9, license code GP9-2314983-RATL-05225; 225 Franklin Street. Fl. 26, Boston, MA, USA 02110; RRID:SCR_002798).

## 4. Conclusions

In this study, two new antibacterial agents possessing antibacterial effects against MDR clinical isolates of Gram-positive species, including ESKAPE bacteria, have been developed. To this end, the synthesis of a double-chain QAS (**6**) was carried out by performing a Wittig reaction using an easily synthesized QPS (**1**). This synthetic strategy allowed us to obtain new representative molecules of both classes of compounds extensively reported to have potent antimicrobial effects on the same synthetic path. QAS was characterized by a quaternary ammonium head and two C14 alkyl chains containing double bonds like those of transfection reagents capable of binding negatively charged counterparts as bacteria membranes. Otherwise, QPS was made of a single C11 alkyl chain and a cationic triphenyl alkyl phosphonium head. Upon complete characterization, the antibacterial activity of **1** and **6** was studied on several multidrug-resistant (MDR) clinical isolates of Gram-positive species, observing remarkable antibacterial activity against strains of the *Enterococcus* and *Staphylococcus* genus. MICs of **1** and **6** were in the range of 4–16 and 4–64 µg/mL, respectively, while 24 h-time-killing experiments carried out with **1** on different MDR *S. aureus* isolates evidenced a bacteriostatic behavior. When used to treat human liver cells HepG2, **1** evidenced good selectivity indices against all isolates considered, while **6** specifically against *E. faecium*. Collectively, while **1** could therefore represent a novel, potent tool for counteracting infections sustained by antibiotic-resistant enterococci and staphylococci bacteria, **6** could be developed as a new therapeutic tool to specifically inhibit *E. faecium*.

## Data Availability

All data supporting the reported results are included in this article and in the related Appendix A. The raw data and datasets are available upon reasonable request.

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
