# Peer review of "Anti Gram-Positive Bacteria Activity of Synthetic Quaternary Ammonium Lipid and Its Precursor Phosphonium Salt"

_ijms, 2024, doi:10.3390/ijms25052761_

Round 1
Reviewer 1 Report
Comments and Suggestions for Authors
In the manuscript entitled Anti Gram-Positive Bacteria Activity of Synthetic Quaternary Lipid and its Precursor Phosphonium Salt Bacchetti et al. report original research describing synthesis and biological activity of new quaternary ammonium derivatives. Upon detailed and complex synthetic route, the Authors assessed the biological activity of two candidates, namely 1 and 6 that were shown to have potent antimicrobial activity against Gram-positive and Gram-negative ATCC bacteria. After antibacterial profiling, detailed activity was further investigated on clinically isolated bacterial strains and was shown that candidates have good bacteriostatic and bactericidal potential. Experiments performed on HepG2 hepatic cell lines, underlined that both candidates have potential for clinical application due to favorable SI values.
The study is well planned in the synthetic part of investigations, but activity study and cytotoxicity measurements are not fully mechanistically explained. However, the presented results are relevant contribution for wider scientific audience, especially for scientist in the field of Medicinal Chemistry and I would therefore recommend the manuscript for publication, but after addressing following points:
Major points:
1. The authors have not compared antibacterial activity of 1 and 6 (MIC) with commercial QACs standards such as BAC and/or CPC. I would recommend either integration of these results from other Authors in the discussion section or independent determination by the Authors. Just comparison with reference antibiotics (and not stating which antibiotics were used) is not comparable or relevant in terms of new QACs synthesis.
2. In the part of describing Potentiometric Titration, it is not clear why did Authors perform these experiments and what was the general idea as for what these data were relevant or used in terms of quaternary ammonium compounds?
3. Figure 14 needs to be improved. Time in hours is not clearly depicted on the graph, and there is no need to show each hour. For clarification it would be practical to show every 4 hours.
4. Concentration dependent cytotoxicity assay was performed in order to investigate the influence of the selected compounds on cell viability. The aim of these experiments was to show how toxic are new compounds to human cells and as such, could they be used in clinics as components of antiseptics and/or disinfectants. These experiments are questionable. First, how cells treated with DMSO as control supposed to kill cells, can result in 100% of cell viability? Reading the material and methods part, I realized that on ordinate Authors depict not cell viability, but rather percentage of control. This control can be live cells or cells that are dead (upon DMSO treatment). What is usually used in this experiment are the live cells as MTT is used for their detection. So Authors need to revise this part of the manuscript, both methodology and obtained results.
Author Response
In the manuscript entitled Anti Gram-Positive Bacteria Activity of Synthetic Quaternary Lipid and its Precursor Phosphonium Salt Bacchetti et al. report original research describing synthesis and biological activity of new quaternary ammonium derivatives. Upon detailed and complex synthetic route, the Authors assessed the biological activity of two candidates, namely 1 and 6 that were shown to have potent antimicrobial activity against Gram-positive and Gram-negative ATCC bacteria. After antibacterial profiling, detailed activity was further investigated on clinically isolated bacterial strains and was shown that candidates have good bacteriostatic and bactericidal potential. Experiments performed on HepG2 hepatic cell lines, underlined that both candidates have potential for clinical application due to favorable SI values.
We thank a lot the Reviewer for his/her positive comments. Anyway, we would like to precise that our two compounds are not both quaternary ammonium salts (QASs), but they are one a QAS and the other a quaternary phosphonium salt (QPS). Moreover, we have not used ATCC bacteria, but in all cases, bacteria were multi drug resistant clinical isolates. Finally, although we hoped that our compounds could be active also on Gram-negative bacteria and bactericidal, they demonstrated a potent bacteriostatic activity against clinical isolates of Gram-positive species, obtaining anyway a relevant result, since they make part of ESKAPE bacteria considered by WHO as the most frightening species, due to their tolerance towards the most part of available antibiotics.
The study is well planned in the synthetic part of investigations, but activity study and cytotoxicity measurements are not fully mechanistically explained. However, the presented results are relevant contribution for wider scientific audience, especially for scientist in the field of Medicinal Chemistry and I would therefore recommend the manuscript for publication, but after addressing following points:
Major points:
- The authors have not compared antibacterial activity of 1 and 6 (MIC) with commercial QACs standards such as BAC and/or CPC. I would recommend either integration of these results from other Authors in the discussion section or independent determination by the Authors. Just comparison with reference antibiotics (and not stating which antibiotics were used) is not comparable or relevant in terms of new QACs synthesis.
We thank the Reviewer for his/her comment concerning the antibacterial agents to be used to compare the antibacterial activity of compounds 1 and 6. Anyway, we have not used the suggested molecules because not clinically administrable. Although structural analogues of 1 and 6, benzalkonium chloride (BAC) and cetylpyridinium chloride (CPC) are not antibiotics to cure human infections by oral or other route of administration, but are biocidal disinfectants present in numerous preparations intended for daily use, domestic or at least oral disinfection and not as therapeutics. BAC is corrosive and toxic. Concentrations between 0.1 and 0.5% cause irritation of the ocular mucous membranes, while concentrations higher than 10% are irritating on the skin. BAC can induce collapse, muscle paresis, liver and kidney changes. CPC is widely used as the active ingredient (0.01–1% w/w) or detergent additive (up to 5 mg/L) in personal care products, but not as antibiotic for human administration, and had proved a 120h-EC50 of 0.1759 µg/mL on fishes, thus demonstrating also a strong ecotoxicity. Since the scope of our research was to find new antibacterial devices capable to counteract bacteria resistance and promising for a future clinical use, we have rationally compared the antibacterial effects of compounds 1 and 6 with those of currently available antibiotics commonly administered for counteracting infections sustained by the bacterial species considered. Also, we have not compared the antibacterial activity of our compounds with that of the commercial QASs suggested by the Reviewer because our aim was not to find new not administrable disinfectants, but new possible therapeutics. From this, our decision to compare the antibacterial effects of 1 and 6 with those of currently available therapeutics. Anyway, for more clarity, specifications on this question have been now included in the text. Please, see lines 490-501. However, to also accept the suggestion of the Reviewer, a comparison of the antibacterial activity of our compound 1 with commercial chlorhexidine has been inserted in lines 530-533. Additionally, in the footnotes of Table 3 (revised version) in the original version of our paper, we had already inserted the names of antibiotics used for comparison, i.e. vancomycin and oxacillin (line 508), now inserted also in the text (lines 496-497).
- In the part of describing Potentiometric Titration, it is not clear why did Authors perform these experiments and what was the general idea as for what these data were relevant or used in terms of quaternary ammonium compounds?
The Reviewer is right. The information requested has been inserted in the abstract (lines 23-24), in the introduction (lines 137-140), and in the Results and Discussion section (lines 437-442 and 471-473).
- Figure 14 needs to be improved. Time in hours is not clearly depicted on the graph, and there is no need to show each hour. For clarification it would be practical to show every 4 hours.
Figure 14 of not revised version and now Figure 10 has been improved according to the suggestions of the Reviewer.
- Concentration dependent cytotoxicity assay was performed in order to investigate the influence of the selected compounds on cell viability. The aim of these experiments was to show how toxic are new compounds to human cells and as such, could they be used in clinics as components of antiseptics and/or disinfectants. These experiments are questionable. First, how cells treated with DMSO as control supposed to kill cells, can result in 100% of cell viability? Reading the material and methods part, I realized that on ordinate Authors depict not cell viability, but rather percentage of control. This control can be live cells or cells that are dead (upon DMSO treatment). What is usually used in this experiment are the live cells as MTT is used for their detection. So Authors need to revise this part of the manuscript, both methodology and obtained results.
We thank the Reviewer having raised this issue and we apologise for the inconvenience. Indeed, we always run in parallel, in each experiment, a DMSO cytotoxic curve (1 ul/ml; 2 ul/ml; 5 ul/ml; 20 ul/ml; 50 ul/ml; 100 ul/ml of DMSO) as positive controls (not shown). The control bars (white colour filled bars), reported in the original Figures, referred to the control wells with cells exposed to the higher concentration of DMSO used to dilute the compounds (namely 1µl/ml and 2µl/ml of DMSO for QPS1 and QAS6, respectively). For clarity, we now indicated in the labels of X axes the concentration of DMSO used for this experimental condition (1µl/ml and 2µl/ml). As clearly intelligible from the results, both DMSO concentrations were not able to induce HepG2, cell death if compared to the condition without DMSO (CTRL, negative control). Thank to the observation of the Reviewer, we now introduce an additional bar (grey edge and crossed bar) in the two figures, showing the cell viability in the experimental condition without DMSO (CTRL). According to these changes we also rigorously recalculate the percentage of cell viability of all the experimental conditions versus “No-DMSO” (CTRL). As expected, the results did not change.
Concerning the data expression, we apologise if this was not clearly explained. In the graphs we report the cell viability in terms of percentage versus control conditions (CTRL) that indeed represent 100% of cell viability. The section of method and the two legends to the figure have been now modified accordingly.
Reviewer 2 Report
Comments and Suggestions for Authors
The manuscript ijms-2878814 "Anti Gram-Positive Bacteria Activity of Synthetic Quaternary Lipid and its Precursor Phosphonium Salt" by Alfei and co-workers describes the synthesis of phosphonium and quaternary ammonium salts and the study of their biological activity. The synthesis was confirmed by 1H, 13C, DEPT NMR, and FTIR spectroscopy. The authors obtained interesting synthetic results and SARs, so I think this paper will be of interest to the readers of International Journal of Molecular Sciences. However, I have a few questions and comments:
1) The use of the term "Quaternary Lipid" is unacceptable. I suggest replacing it with "Lipid-Based Quaternary Ammonium Salt" or "Quaternary Ammonium Lipid".
2) The abstract should be shortened by removing synthetic details from other sections of the manuscript. I also recommend avoiding unnecessary abbreviations in the abstract.
3) The authors list a variety of quaternary ammonium salts in the introduction, while they do not write about the promise of QASs with more than two (gemini QASs) positively charged moieties. I recommend the authors to strengthen the Introduction part about multicationic and polyQASs. Recent articles on this topic should be added.
4) The authors need to clearly write which compounds are new and which were previously obtained. It is not necessary to give a detailed description of the spectra of previously obtained compounds. A reference to a publication with a methodology is enough.
5) The manuscript is very long due to the large amount of spectral data. Images of the spectra of the initial compounds can be transferred to the supplementary materials. Schemes 2-7 also duplicate Scheme 1. I propose to delete Schemes 2-7.
6) Is compound 6 a yellow oil (line 371) or red-orange solid (line 765)? Generally, chloride ammonium salts are white or pale yellow powders.
7) In general, amines and quaternary ammonium salts tend to chromatograph very poorly on silica gel. Were there any problems in this case?
8) It is not clear from the text of the manuscript why the potentiometric titration was done. Please specify.
9) Tables 1 and 2. Please add commercially available positively charged antibiotics (like benzalkonium chloride, chlorhexidine, etc.) as control. It is also necessary to add a control for experiments with healthy cells and compare selectivity indices.
10) Minor corrections:
- Line 35, "benzalkonium hydrochloride" should be "benzalkonium chloride".
- "in vivo" should be in italic.
- Data cannot be found for some references, e.g., [18] and [19].
- The integral intensities in NMR spectra should correspond to the number of protons in the molecule. Please correct this.
- Line 204, "alkene 2" is not alkene, compound 2 is alken-1-ol (enol). Line 236, "vinyl alcohol 2" is also incorrect.
- Line 749, "the chloroform (CH3Cl)". Please correct to CHCl3.
- Figure S12. Please correct the formula of compound 4.
- Please add a scale bar (line) to all microscopy images (Figures 15 and 16).
Author Response
The manuscript ijms-2878814 "Anti Gram-Positive Bacteria Activity of Synthetic Quaternary Lipid and its Precursor Phosphonium Salt" by Alfei and co-workers describes the synthesis of phosphonium and quaternary ammonium salts and the study of their biological activity. The synthesis was confirmed by 1H, 13C, DEPT NMR, and FTIR spectroscopy. The authors obtained interesting synthetic results and SARs, so I think this paper will be of interest to the readers of International Journal of Molecular Sciences. However, I have a few questions and comments:
1) The use of the term "Quaternary Lipid" is unacceptable. I suggest replacing it with "Lipid-Based Quaternary Ammonium Salt" or "Quaternary Ammonium Lipid".
We thank a lot the Reviewer for his/her comment. Accordingly, we have replaced the incorrect expression with those suggested. Please, see lines 3, 13 and 34.
2) The abstract should be shortened by removing synthetic details from other sections of the manuscript. I also recommend avoiding unnecessary abbreviations in the abstract.
According to the Reviewer suggestions, the abstract has been shortened, synthetic detail have been removed and abbreviations have been avoided. Specifically, see deletions at lines 17-21.
3) The authors list a variety of quaternary ammonium salts in the introduction, while they do not write about the promise of QASs with more than two (gemini QASs) positively charged moieties. I recommend the authors to strengthen the Introduction part about multicationic and polyQASs. Recent articles on this topic should be added.
We apologise to the Reviewer for our forgetfulness, and we thank him/her for the note. Thanks to his/her suggestion we have now included in the Introduction section the results recently reported in literature obtained with multi QACs on different bacterial species (lines 71-75 and 83-107). Five additional references [16-20] have been added. References list has been updated.
4) The authors need to clearly write which compounds are new and which were previously obtained. It is not necessary to give a detailed description of the spectra of previously obtained compounds. A reference to a publication with a methodology is enough.
As reported in the main text, the synthetic process from 11-bromo-undecan-1-ol to compound 6 reported in this paper has been described in a Doctoral Thesis, but no publication exists reporting all the procedure up to compound 6. Otherwise, the synthesis of compounds 1 and 2 have been previously reported (Ref 30 in the revised manuscript), but only a mere peak list for both FTIR and 1H NMR has been provided, and no discussion has been given. Additionally, in Ref 30, 13C NMR is missing for both compounds and no image is available. So, we thought that a more careful description of spectral data of all intermediates and final compounds would be necessary.
5) The manuscript is very long due to the large amount of spectral data. Images of the spectra of the initial compounds can be transferred to the supplementary materials. Schemes 2-7 also duplicate Scheme 1. I propose to delete Schemes 2-7.
As suggested by the Reviewer, Scheme 2-7 have been removed from the main text and inserted in Supplementary Materials, as well as Figures 6-9. The Figure numbering has been updated, both in the main text and in the Supplementary Materials, as well as the Table of Contents of Supplementary Materials.
6) Is compound 6 a yellow oil (line 371) or red-orange solid (line 765)? Generally, chloride ammonium salts are white or pale yellow powders.
We apologise to the Reviewer for our error. Compound 6 was a yellow oil with tendency to solidify. The point has been corrected (line 803).
7) In general, amines and quaternary ammonium salts tend to chromatograph very poorly on silica gel. Were there any problems in this case?
Except for the modest reaction yields (65 and 52%), evidencing that some material was retained into the column, no particular problem was evidenced during elution, mainly thanks to the presence of the two long alkenyl chains, which conferred the compounds the sufficient hydrophobicity to make the chromatographic purification easier.
8) It is not clear from the text of the manuscript why the potentiometric titration was done. Please specify.
The Reviewer is right. We have omitted necessary information concerning why we have performed potentiometric titrations of 1 and 6. The information requested has been inserted in the abstract (lines 23-24), in the introduction (lines 137-140), and in the Results and Discussion section (lines 437-442 and 471-473).
9) Tables 1 and 2. Please add commercially available positively charged antibiotics (like benzalkonium chloride, chlorhexidine, etc.) as control. It is also necessary to add a control for experiments with healthy cells and compare selectivity indices.
We thank the Reviewer for his/her comment concerning the reference compounds to be inserted in Table 2 and 3 (revised and corrected version). Anyway, we have not used the suggested molecules because not clinically administrable. Benzalkonium chloride (BAC), chlorhexidine (CHX), as well as cetylpyridinium chloride (CPC) are not antibiotics to cure human infections by oral or other route of administration, but are biocidal disinfectants present in numerous preparations intended for daily use, domestic or at least oral disinfection and not as therapeutics. BAC is corrosive and toxic. Concentrations between 0.1 and 0.5% cause irritation of the ocular mucous membranes, while concentrations higher than 10% are irritating on the skin. BAC can induce collapse, muscle paresis, liver and kidney changes. CPC is widely used as the active ingredient (0.01–1% w/w) or detergent additive (up to 5 mg/L) in personal care products, but not as antibiotic for human administration, and had proved a 120h-EC50 of 0.1759 µg/mL on fishes, thus demonstrating also a strong ecotoxicity. CHX is harmful if ingested and irritates the skin and eyes. Since the scope of our research was to find new antibacterial devices capable to counteract bacteria resistance and promising for a future clinical use, we have rationally compared the antibacterial effects of compounds 1 and 6 with those of currently available antibiotics commonly administered for counteracting human infections sustained by the bacterial species considered. Also, we have not compared the antibacterial activity of our compounds with that of the commercial QASs suggested by the Reviewer, because our aim was not to find new not administrable disinfectants, but new possible therapeutics. From this, our decision to compare the antibacterial effects of 1 and 6 with those of currently available therapeutics. Anyway, for more clarity, specifications on this question have been now included in the text. Please see lines 490-501. However, to also accept the suggestion of the Reviewer, a comparison of the antibacterial activity of our compound 1 with commercial chlorhexidine has been inserted in lines 530-533. As for the other request of the Reviewer to test also the reference antibiotics used on healthy cells to compare their SI with the SI of our compounds, having we selected antibiotics clinically approved, in our opinion, it could be a questionable experiment. In fact, since vancomycin and oxacillin have been approved for clinical used from year it is logical that they have wide therapeutic windows verified both in vitro and in vivo.
10) Minor corrections:
- Line 35, "benzalkonium hydrochloride" should be "benzalkonium chloride".
Corrected (lines 38-39).
- "in vivo" should be in italic.
Corrected.
- Data cannot be found for some references, e.g., [18] and [19].
Ref 19 of the original paper (not revised) has been removed, while Ref 18 of the original paper, and now Ref 25, as specified in the text and above reported (point 4) is a not published Doctoral Thesis. Unpublished materials not intended for publication are allowed by IJMS as reported in the instruction for the authors. Please see at the following link:
https://www.mdpi.com/journal/ijms/instructions
- The integral intensities in NMR spectra should correspond to the number of protons in the molecule. Please correct this.
On the Reviewer suggestion, we have tried to improve discrepancies where possible, but to excessive force the integral provided by the instrument, in our opinion, is not scientifically correct. Anyway, after having carefully checked the correspondence between the integrals’ intensities and the number of protons in the 1H NMR spectra present in the paper, we have found that for the most part of peaks a perfect correspondence exists, while for few peaks we have detected minimal differences, that anyway are acceptable especially in molecules having peaks standing for many proton atoms and difficult to be integrated correctly.
- Line 204, "alkene 2" is not alkene, compound 2 is alken-1-ol (enol). Line 236, "vinyl alcohol 2" is also incorrect.
Corrected (lines 237, 238-239 and 270).
- Line 749, "the chloroform (CH3Cl)". Please correct to CHCl3.
Corrected (line 786).
- Figure S12. Please correct the formula of compound 4.
As asked, the formula of compound 4, in Figure S12 of the original manuscript, now Figure S14 has been corrected. We thank the Reviewer for his/her valuable contribution.
- Please add a scale bar (line) to all microscopy images (Figures 15 and 16).
We apologise for this inattention, the scale bars in Figure 15 and 16 (original paper), now Figure 11 and 12 have been introduced in all microscopy images.
Round 2
Reviewer 1 Report
Comments and Suggestions for Authors Dear authors, I am satisfied with the accepted comments in order to improve your manuscript and I consider the full study, including results and interpretation is now ready for the publication.Regards,
Reviewer 2 Report
Comments and Suggestions for Authors
I thank the authors for answering my questions and improving the manuscript.
Round 3
Reviewer 2 Report
Comments and Suggestions for Authors
I thank the authors for answering my questions and improving the manuscript.